# Alterations in Anandamide Synthesis and Degradation during Osteoarthritis Progression in an Animal Model

**DOI:** 10.3390/ijms21197381

**Published:** 2020-10-06

**Authors:** Marta Bryk, Jakub Chwastek, Magdalena Kostrzewa, Jakub Mlost, Aleksandra Pędracka, Katarzyna Starowicz

**Affiliations:** Maj Institute of Pharmacology, Polish Academy of Sciences, 31-343 Cracow, Poland; bryk@if-pan.krakow.pl (M.B.); chwastek@if-pan.krakow.pl (J.C.); m.kostrzewka@gmail.com (M.K.); mlost@if-pan.krakow.pl (J.M.); o.pedracka@gmail.com (A.P.)

**Keywords:** osteoarthritis, endocannabinoid system, endocannabinoids, pain, chronic pain, TRPV1, animal model, cartilage, synovial membrane, spinal cord

## Abstract

Osteoarthritis (OA) is a degenerative joint disease manifested by movement limitations and chronic pain. Endocannabinoid system (ECS) may modulate nociception via cannabinoid and TRPV1 receptors. The purpose of our study was to examine alterations in the spinal and joint endocannabinoid system during pain development in an animal model of OA. Wistar rats received intra-articular injection of 3mg of sodium monoiodoacetate (MIA) into the knee joint. Animals were sacrificed on day 2, 7, 14, 21, 28 after injection and lumbar spinal cord, cartilage and synovium were collected. Changes in the transcription levels of the ECS elements were measured. At the spinal level, gene expression levels of the cannabinoid and TRPV1 receptors as well as enzymes involved in anandamide synthesis and degradation were elevated in the advanced OA phase. In the joint, an important role of the synovium was demonstrated, since cartilage degeneration resulted in attenuation of the changes in the gene expression. Enzymes responsible for anandamide synthesis and degradation were upregulated particularly in the early stages of OA, presumably in response to early local joint inflammation. The presented study provides missing information about the MIA-induced OA model and encourages the development of a therapy focused on the molecular role of ECS.

## 1. Introduction

Osteoarthritis (OA) is a whole joint disease characterized by cartilage destruction, subchondral bone alterations and synovial membrane inflammation. Currently, OA is a leading cause of chronic pain and disability worldwide [1]. Current treatment is very limited and is mainly based on symptomatic pain relief by nonsteroidal anti-inflammatory drugs (NSAIDs) offering insufficient pain relief [2,3]. Despite the research progress, OA remains among the most challenging joint diseases due to the lack of self-healing capacity of articular cartilage and there is no cure for it. Therefore, an extensive search for new treatment options is needed. Preclinical studies suggest, that endocannabinoids have a potential to become novel molecular targets for drug development in chronic pain, including pain resulting from osteoarthritis [4].

The endocannabinoid system (ECS) is composed of cannabinoid receptors, CB1 and CB2, their endogenous ligands and their synthetic and metabolic enzymes. Anandamide (AEA) is the first identified and best characterized endogenous ligand of cannabinoid receptors [5]. Levels of AEA are altered in the spinal cord of neuropathic and osteoarthritic animals [6,7], which emphasizes the importance of AEA in nociceptive processing in chronic pain. Valastro et al. demonstrated that AEA is present in the synovial fluid of arthritic knees in dogs [8]. Moreover, an increase in the level of AEA was detected in the synovial fluid collected from the patients with OA compared with that in the healthy controls [9]. Unlike conventional neurotransmitters, endocannabinoids are not stored but are locally synthesized on-demand in the areas of cellular stress. The main AEA synthesis pathway includes phospholipid precursor *N*-arachidonoyl phosphatidylethanolamine (NAPE), which is hydrolyzed by *N*-arachidonoyl phosphatidylethanolamine phospholipase D (NAPE-PLD) in the presence of Ca^2+^ [10]. However, there are two additional well-characterized pathways associated with AEA production in a Ca^2+^-independent manner. The phospholipase C (PLC) pathway includes PLC and two other concurrent enzymes, protein tyrosine phosphatase non-receptor type 22 (PTPN22) and phosphatidylinositol-3,4,5-trisphosphate 5-phosphatase 1 (INPP5D) [11,12]. Another alternative synthesis pathway involves secreted phospholipase A2 (sPLA2), β hydrolase domain-containing protein 4 (ABHD4) and glycerophosphodiester phosphodiesterase 1 (GDE1) [13]. AEA is also characterized by a short half-life particularly due to an effective enzymatic degradation by three enzymatic pathways, including fatty acid amide hydrolase (FAAH) generating arachidonic acid (AA) and ethanolamine (ETA); cyclooxygenase 2 (COX-2) generating prostaglandins (PG) and prostamides (PM); and arachidonate lipoxygenases 12 and 15 (LOX-12/15) generating 12/15 hydroxyeicosatetraenoylethanolamide (12/15-HETE-EA), respectively (Figure 1) [7,10,14].

AEA degradation reduces its use for analgesic proposes; therefore, targeting (by pharmacological or genetic inhibition) its degrading enzymes is a promising strategy to treat the pain syndromes. However, AEA may interact and activate other targets, such as the transient receptor potential vanilloid type 1 (TRPV1) [15]. TRPV1 was originally reported to be expressed on the peripheral nociceptive afferents and is present in the spinal and supraspinal structures [16]; TRPV1 is also expressed in multiple cell types of the joint [17,18,19]. Interestingly, a variant of the TRPV1 gene is associated with the risk of the development of symptomatic OA [20]. Other studies indicate that inflammation and neuronal injury alter the localization of TRPV1 and sensitize the channel to respond to stimulus modalities beyond the classical thermal profile [21]. TRPV1 contains multiple phosphorylation sites; hence, activation by kinases, such as Ca2+/calmodulin-dependent protein kinase (CaMKII), is one of the proposed mechanism of activation and sensitization of TRPV1 [22,23].

Unrelenting pain can induce changes in the spinal cord, which play an essential role in the integration and modulation of nociceptive signals from the peripheral tissues to the higher centers of the brain. An emerging concept is that the expression of CB1 and TRPV1 in the same or in neighboring cells, especially in the case of activation by the same mediator, such as AEA, enables a cross-talk with variable impact on pain perception. Thus, we aimed to improve the understanding of a possible cross-talk between the endocannabinoid and endovanilloid systems to assist with fine-tuning of analgesic strategies particularly for patients suffering from chronic pain. The degree of pain does not always correlate with the extent of joint damage or presence of active inflammation, we investigated the occurrence of a central component of OA pain by correlating OA pain behavior of the animals reported in the previous studies [24] with the assessment of ATF-3, a marker of spinal neuronal activity. Involvement of mitogen-activated protein kinases 3 and 14 (MAPK3 and MAPK14, respectively) in the chronic pain formation has been shown in preclinical studies [25,26]. Furthermore, the present investigation aimed to examine the spinal and joint endocannabinoid systems in combination with TRPV1 and its sensitization factors during pain development in a rat sodium monoiodoacetate (MIA) model of OA. We examined changes in the transcription of the ECS system elements, including cannabinoid receptors and multiple AEA synthesis and degradation enzymes, in the lumbar spinal cord, cartilage and synovial membrane of osteoarthritic rats. Inflammation and tissue hypoxia associated with OA lead to a decrease in local tissue pH resulting in TRPV1 sensitization [22,27]. This decrease may contribute to pain development; however, the downstream signaling of this pathway has been insufficiently characterized. Thus, we also investigated changes in the mRNA expression of TRPV1 receptor and CaMKII, which is involved in TRPV1 sensitization during OA development. In summary, we aimed to increase our understanding of the role of the endocannabinoid system in the pathogenesis of OA and to propose new molecular targets and therapeutic strategies for OA treatment to increase the efficacy of OA pain management.

## 2. Results

During OA development, a significant elevation of mRNA level of a neuronal marker of nerve injury *Atf3* in the dorsal lumbar spinal cord was detected 28 days after MIA injection (Figure 2A). MAP kinases p38 (*Mapk14*) and ERK1 (*Mapk3*) were detected in the L4- L6 spinal cord segments after MIA injection. Analysis of mRNA abundance revealed a significant increase in *Mapk3* exclusively 28 days after MIA treatment (Figure 2B). *Mapk14* gene was strongly upregulated on days 21 and 28 (Figure 2C). In the cartilage OA samples, no significant changes in the *Mapk3* and *Mapk14* gene expression levels were detected (Figure 2D,E). In the synovial membrane samples collected from OA rats, the levels of *Mapk3* and *Mapk14* were elevated only 7 days after MIA treatment (Figure 2F,G).

### 2.1. Changes in the Cnr1, Cnr2 and Trpv1 Gene Expression in the Dorsal Lumbar Spinal Cord and Joint Tissue of Osteoarthritic Rats

In rats with developed OA, a different pattern of *Cnr1* and *Cnr2* gene expression (encoding CB1 or CB2 receptors, respectively) was observed in the dorsal lumbar L4-L6 spinal cord segments during the development of OA pain. A significant increase in the *Cnr1* gene expression was detected on day 28 after MIA injection in the ipsilateral part of the spinal cord (Figure 3A). An increase in the *Cnr2* transcript was observed on day 7 and the levels of the transcript were decreased at later time points (Figure 3B). The *Trpv1* mRNA level was significantly elevated exclusively on day 28 (Figure 3C).

In the OA cartilage samples, no significant changes were detected in the *Crn1* and *Cnr2* gene expression (Figure 3D,E), whereas a significant elevation in the *Trpv1* gene expression was observed on day 21 after MIA injection (Figure 3F).

In the synovial membrane collected from OA rats, the *Cnr1* gene expression was below the detection limit (Figure 3G); however, the *Cnr2* gene expression was increased starting from day 2 after MIA injection and was significantly elevated starting from day 14 till the end of the experiment (Figure 3H). The expression level of *Trpv1* has increased only 14 days after MIA injection (Figure 3I).

### 2.2. Expression of the Main Enzymes of AEA Synthesis and Degradation in the Dorsal Lumbar Spinal Cord and Joint Tissue of MIA-Treated Rats

The analysis of the transcript levels of the enzymes of main AEA synthesis and degradation, including *Nape-pld* and *Faah*, in the dorsal lumbar spinal cord during OA progression indicated an incremental increase along with the disease progression. No significant changes were observed in the *Nape-pld* expression on days 2, 7, 14 and 21 after MIA injection. Substantial *Nape-pld* upregulation was detected only on day 28 (Figure 4A). The *Faah* expression showed a trend to gradually increase from day 7 and was significantly elevated on days 21 and 28 (Figure 4B).

No significant changes were detected in the cartilage of MIA-treated rats (Figure 4C,D).

An increase in the *Nape-pld* gene expression in the synovial membrane samples was observed two days after MIA injection and persisted until the end of the experiment (Figure 4E). A rising trend in the *Faah* gene expression was observed on day 21; however, the results did not reach statistical significance (Figure 4F).

### 2.3. Alterations in the Gene Expression of the Alternative AEA Synthesis and Degradation Pathways in the Lumbar Spinal Cord and Joint Tissue of Rats after MIA Injection

OA caused by intra-articular (i.a.) 3 mg MIA injection leads to the changes of the levels of mRNA encoding the enzymes of the alternative AEA synthesis and degradation pathway in the dorsal lumbar L4-L6 spinal cord segments, cartilage and synovial membrane. In the lumbar spinal cord, the gene expression levels of phospholipase C (*Plc*) and protein tyrosine phosphatase non-receptor type 22 (*Ptpn22*) tended to increase in response to MIA injection; the highest significant expression was detected 28 days after OA induction in the spinal cord (Figure 5A,B). Phosphatidylinositol-3,4,5-trisphosphate 5-phosphatase 1 (*Inpp5d*) was increased on day 7 and the increase persisted till day 28 (Figure 5C). Phospholipase A2 (*sPla2*) mRNA level was upregulated starting from day 7 after MIA injection; however, the highest expression was observed on days 14 and 21 (Figure 5D). The level of β hydrolase domain-containing protein 4 (*Abdh4*) mRNA was elevated only on day 28 (Figure 5E). The expression of the glycerophosphodiester phosphodiesterase 1 (*Gde1*) transcript was significantly increased 2 and 28 days after MIA injection (Figure 5F). MIA injection increased the levels of cyclooxygenase 2 (COX-2 encoded by the *Ptgs2* gene) and arachidonate lipoxygenases 12 (*Alox12*) transcripts on day 28 (Figure 5G,H). The expression of arachidonate lipoxygenases 15 (*Alox15*) had a trend to increase at the last two time points; however, no significant changes were observed (Figure 5I).

In the cartilage samples, an increase in the gene expression was observed only for *Plc* on day 14 (Figure 6A), *Ptpn22* on day 7 (Figure 6B) and *Ptgs2* on days 2-21 with a significant increase on day 14 (Figure 6G). A time-dependent decrease in the *sPla2* gene expression was observed (Figure 6D).

OA caused by MIA injection resulted in certain changes in the expression of the synovial membrane genes involved in the alternative AEA synthesis and degradation pathways. Expression of *Ptpn22* gene was significantly increased starting from day 2 after MIA treatment and remained increased until the end of the experiment (Figure 7B). The *Inpp5d* gene expression was elevated shortly after MIA injection (days 2 and 7) and was gradually decreased till day 28, when it almost reached a baseline (Figure 7C). A significant decrease in the *sPla2* gene expression was detected throughout the entire experiment (Figure 7D). The *Abdh4* and *Gde1* gene expression levels were gradually increased from the beginning of the experiment, reaching the highest level in the middle (day 7 of 14 days, respectively). Subsequently, the level gradually decreased until the end of the experiment (Figure 7E,F). The level of the *Alox12* gene expression was elevated from day 7 to the end of the experiment (Figure 7H). Changes in other genes (*Plc*, *Ptgs2*, *Alox15*) did not reach statistical significance.

### 2.4. Overexpression of Calcium/Calmodulin-dependent Protein Kinase II Delta (CaMKII) as a Result of MIA-Induced OA-like Changes

TRPV1 sensitization may be involved in chronic pain caused by osteoarthritis. Transcript of CaMKII, a kinase which is involved in TRPV1 activation, was detected in the L4 L6 spinal cord segments of the rats. Analysis of mRNA levels indicated a significant increase in *Camk2d* on days 7, 21 and 28 after MIA injection (Figure 8A). In the OA cartilage samples, a significant increase in the *Camk2d* gene expression was detected 14 and 21 days after MIA injection (Figure 8B). In the synovial membranes collected from OA rats, the mRNA level of *Camk2d* were increased in the middle of the experiment, 7 and 14 days after MIA treatment (Figure 8C).

## 3. Discussion

Changes associated with the late stages of OA have been extensively studied; however, the exact mechanism of OA development and the role of the ECS molecules are poorly understood. The present study confirms the changes in the expression of the ECS molecules during OA development. The data indicate that the mRNA levels of cannabinoid receptors (CB1 and CB2) are elevated in the spinal cord or synovial membrane, respectively, which may indicate a role of endocannabinoids in OA progression. Moreover, the changes in the expression of the genes encoding for the enzymes participating in various AEA metabolic pathways were detected, which may play a role in OA pain development.

In the current study, the connection between MIA-induced osteoarthritic pain and central nerve sensitization is demonstrated. The gene expression of *Atf3* (a neuronal marker of nerve injury) is proven to be altered in the dorsal root ganglia; however, the changes are also present in the spinal cord after nerve injury or in chronic pain [28,29,30]. *Atf3* gene was significantly increased in the advanced OA stage, indicating a neuronal alteration at the spinal level in the advanced OA stages. In the neuropathic pain model (ligation of the L5 spinal nerve), Jin et al. demonstrated that the lesions result in immediate phosphorylation of Mapk14 (p-p38 increase) in the ipsilateral spinal cord microglia and delayed activation of Mapk14 in the L5 DRG neurons [31]. Mapk14 is involved in TRPV1 sensitization in DRGs via inflammation-induced NGF upregulation. Under the inflammatory conditions, NGF in DRGs contributes to Mapk14 phosphorylation, which increases TRPV1 translation and transport to the peripheral terminals. This process entails an increase in the pain sensitivity [26]. Moreover, in the primary DRG culture, ERK (MAPK3) participated in the sensitization of TRPV1 during inflammation [32]. Our data indicate the importance of *Mapk3* and *Mapk14* genes at the later OA stage, when chronic pain is fully developed.

Preclinical and clinical studies have suggested that the endocannabinoid system may be useful to treat various diseases, including the diseases related to chronic pain. In the course of several diseases, an elevated level of the ECS components was reported in preclinical models [33] and in clinical trials [34,35]. Bishay et al. demonstrated that aged mice have lower AEA levels in particular brain structures compared with that in young mice, which resulted in stronger nociceptive development after spared nerve injury and weaker response for R-flurbiprofen [36]. In particular, an increase in the level of ECS molecules was observed during OA progression at the local [9] and spinal levels [37]. A number of studies emphasized the role of ECS modulators in the alleviation of pain in animal OA models [38,39,40,41].

Moreover, on the spinal level, in the advanced phase of the disease, significant alterations in the expression levels of several investigated genes were detected. From the therapeutic point of view, enzymes of the AEA metabolic pathway are more important than receptors on which AEA acts. The main enzymes responsible for AEA synthesis and degradation (*Nape-pld* and *Faah*) showed a similar pattern of expression with an increase at the end stages of the experiment. Moreover, the enzymes involved in the alternative pathways of AEA synthesis and degradation (*Plc*, *Ptpn22*, *Inpp5d*, *sPla2*, *Abdh4*, *Gde1*, *Ptgs2*, *Alox12* and *Alox15*) showed a trend to increase after MIA injection, particularly at the later stages of OA. Interestingly, similar patterns of gene changes were observed in the previous studies. Malek et al. investigated changes in the genes in the DRGs and spinal cord of neuropathic rats. The results indicated significant upregulation of several ECS member genes after CCI induction, including *Crn1*, *Cnr2*, *Pla2g2a*, *Plcb1*, *Inpp5d*, *Faah*, *Ptgs2*, *Alox-12* and *Alox-15* [42]. *Alox-15* gene expression and protein synthesis in the lumbar spinal cord were confirmed in a CCI model already 7 days after the surgery [7]. Guo et al. showed an increase in CAMKII, TRPV1 and pERK1/2 protein levels in the spinal dorsal horn of animals suffering from chronic pain. Moreover, silencing of the TRPV1 gene attenuated mechanical and thermal hyperalgesia and CaMKII and ERK2 phosphorylation [25]. On the other hand, CaMKII is required for TRPV1 ligand binding. Only phosphorylated TRPV1 can be activated by capsaicin and mutations in the TRPV1 at CaMKII sites fail to elicit capsaicin-sensitive currents [23]. In our study, a significant enhancement of the *Camk2d* gene expression in the lumbar spinal cord was observed, particularly in the advanced phase of OA, which may be associated with TRPV1 sensitization in the advanced OA stage.

The data of the present study indicate that NAPE-PLD (the main enzyme responsible for AEA synthesis) plays a role during chronic pain formation; additionally, several alternative factors are important for nerve sensitization in OA progression. Similarly, there is more than a single AEA degradation pathway. Our study revealed that alternative enzymes involved in AEA degradation are upregulated under the OA conditions on the spinal level. This result provides insight into the spinal nociceptive processing and central sensitization during OA development. Important, the majority of the genes were upregulated on the spinal level in the advanced phase of the disease (14 days after MIA injection). This result indicates that ECS molecules may modulate spinal pain processing during the advanced phase of the disease. MIA-induced OA model requires several days to develop the changes, which cause chronic pain, in the cartilage, synovium and subchondral bone. Studies using animal OA models demonstrate constant animal pain responses several days after MIA-injection. In the previous studies, a precise behavioral description of the changes in pain threshold and molecular alterations in DRGs after 3 mg MIA injection was provided. A biphasic pain response was observed in the early inflammatory stage (associated with i.a. injection) and advanced stage beginning on day 14 after MIA treatment [24]. Thus, in most experiments, days 21 or 28 after i.a. MIA injection are used for drug testing. Nevertheless, spinal OA-related alterations arise from the chronic pain impulses transmitted from the animal joints to the nervous system. Sagar et al. demonstrated that in advanced OA phase (28-31 days after OA induction), the response of the wide dynamic range (WDR) dorsal horn neurons induced by innocuous and noxious mechanical stimulation of the hind paw was significantly increased in MIA treated animals compared with that in the control group [37]. Thus, ECS molecules may play an important role especially in the advanced phase of OA on the spinal level.

In addition to the central sensitization, measured at the level of the lumbar spinal cord, local changes in ECS molecules were investigated in our study. Cartilage is the main joint tissue, which covers bone surfaces, prevents bone abrasion and facilitates joint movements. During OA development, degenerated cartilage fragments released into the joint space sensitize the synovial membrane. Thus, the secretion of proinflammatory factors and local inflammation (synovitis) are induced. The exact mechanism of this process remains unknown (especially for idiopathic OA); however, disease progression represents a self-perpetuating cycle of inflammation and extracellular matrix degradation, in which the main role is played by cartilage, synovial membrane and subchondral bone [43]. The main degenerating tissue during OA is cartilage; however, the malfunctioning of the synovial membrane also has very serious consequences. The synovial membrane is responsible for the production of the synovial fluid and joint lubrication; hence, it plays a primary secretory role in the joint. Thus, the synovial membrane can be more sensitive to gene expression alterations during OA. In the present study, higher variability in the gene expression in the synovial membrane was detected compared with that in the cartilage of rats after MIA treatment.

In the cartilage samples, our results indicate a less significant role of cartilage in ECS functioning during OA progression as the advanced OA phase was not accompanied by significant ECS alterations. However, the cartilage tissue is the most vulnerable tissue to degeneration during the disease progression. Histological staining demonstrated that in 1 mg MIA-induced OA model after 14 days, cartilage is significantly degraded; however, starting from 21 day after MIA injection, the cartilage is essentially completely destroyed (Appendix A). Based on the 3Rs approach to animal testing, testing of the 3 mg dose was not performed when the advanced cartilage changes were observed after administration of 1 mg of MIA. This phenomenon may explain less significant role of cartilage in ECS functioning during OA progression. Alternative pathways (including PLC, PTPN22 and COX-2) may be involved in the early stage of the disease and transition from the early to advanced stage, since alterations in the expression of these genes were observed. However, a more thorough study is needed to explain and clarify these changes.

The role of the synovial membrane was investigated in detail. A significant increase in the CB2 receptor gene expression was detected during almost the entire experimental period. The mRNA of the TRPV1 receptor reached a maximum on day 14 suggesting the main role of the CB2 receptor, rather than TRPV1, in chronic pain associated with OA, with marginal role of CB1 (not detectable, in agreement with [44]). The results of the analysis of the genes involved in TRPV1 sensitization confirm more important role of CB2 rather than TRPV1 in advanced OA phase, including an increase in the mRNA of the *Camk2d* gene in the middle of the study, which returned to the baseline at the end of the experimental period. This result indicates that TRPV1 did not play a very important role in advanced OA in an animal model. The presented study is restricted in targeting of this type of receptor during clinical trials in patients with advanced stage OA. Several clinical studies with TRPV1 inhibitors had been conducted; however, the results have been poor [45].

Furthermore, based on our results an important role of the main AEA synthesis pathway in the synovial membrane at the early stages of OA is suggested. This result is consistent with the data of the literature that elevated level of AEA was detected in the synovial tissue and fluid of patients suffering from OA [9]. Our results indicate that degrading enzymes apparently do not play a key role in OA progression. Nevertheless, these genes cannot be completely skipped when planning the treatment strategies. Similarly to already discussed spinal cord and cartilage, significant increase in synovial’s expression levels of genes involved in the alternative AEA synthesis and degradation pathways (*Ptpn22*, *Alox12*) was observed at the latter stages of OA. *Inpp5d*, *Abdh4* and *Gde1* gene expression was augmented in the initial phase, while *sPla2* was diminished since the 2^nd^ day of the study similarly to the pattern observed in the cartilage samples. This result indicates that separate sets of genes may be important in particular OA stages. Considerable changes have been observed in the present study; however, additional investigation is required to draw firm conclusions about the role of the ECS molecules on the local level (in cartilage and synovial membrane) during the progression of OA.

In the current study we proved the AEA synthesis and degradation enzyme upregulation, during the course of OA. Increase in the AEA synthesis enzymes lead to an increase in the AEA level, which is a desired result, because of the AEA’s analgesic effect. In turn, AEA degradation enzymes’ enhancement increase the level of AEA metabolites (arachidonic acid, ethanolamine, prostaglandins, prostamides, 12-/15-HETE-EA) and reduce the level of AEA. The latter AEA’s metabolites, derived on the main and alternative pathways can be involved in the inflammatory process. Therefore an effective approach to omit problem might be dual-acting substances, for example, FAAH/COX-2 inhibitors, that target both enzymes. Dual-acting drugs offer an analgesic effect by elevating the endogenously produced endocannabinoids (by inhibiting FAAH) and lower the production of pro-inflammatory prostaglandins (by inhibiting COX-2) [46]. In turn, LOX-12/15 metabolites may act in an analgesic way. Indeed, in an animal model of neuropathic pain (chronic constriction injury, CCI, to the sciatic nerve), FAAH inhibitor URB597, diminished thermal and tactile allodynia but also decreased the spinal AEA level and increased LOX-15 level at the same time. This may lead to the TRPV1-mediated analgesia in CCI rats, via 15-hydroxy-AEA, together with oleoylethanolamide and palmitoylethanolamide [7]. AEA metabolites can also be important in several other pathologies. Turcotte et al. widely summarizes the regulatory role of AEA metabolites in various diseases [47]. Nevertheless, to confirm this hypothesis the levels of metabolites should be measured in the animals’ tissues, what is a proper direction for the future research.

Considering the analgesic and anti-inflammatory effects of cannabinoids in the pre-clinical studies, cannabinoid therapy seems to be a promising target for the treatment of several diseases. In the arthritis animal models, phytocannabinoid cannabidiol (CBD) reduced inflammation and analgesia in a rat model [48,49]. CBD may also preferentially target inflammatory-activated fibroblasts and reduce its viability, therefore may have an anti-arthritic activity [50]. Synthetic CB2 receptor agonists were also proven to exert an anti-inflammatory response in several arthritis animal models. JWH-015 inhibited inflammation in the rheumatoid arthritis synovial fibroblasts cell cultures and in the arthritis rat model [51]. JWH133 suppressed collagen-induced arthritis in mice, acted anti-inflammatory by repolarizing macrophages from the M1 to M2 phenotype and reduced pro-inflammatory cytokine expression [52]. CB2 agonist 4Q3C showed an anti-inflammatory effect in rheumatoid arthritis mouse model (reduced bone erosion, inhibited formation of osteoclasts and lowered the level of TNFα, IL-1β, COX-2 and inducible NO synthase) [53].

## 4. Materials and Methods 

### 4.1. Animals

All experiments were approved by the Local Bioethics Committee of the Institute of Pharmacology (Cracow, Poland, approval numbers: 938/2012; 125/2018). Male Wistar rats (Charles River, Hamburg, Germany) initially weighing 225–250 g were used for all experiments. Animals were housed 5 per cage under a 12/12-h light/dark cycle with food and water available *ad libitum*. All experiments were performed in the morning hours (between 9:00 and 12:00). Experimental groups consisted of *n* = 9 animals (for spinal cord isolation) or *n* = 6 animals (for cartilage/synovium isolation). In rare cases, individual samples had to be excluded from the analysis due to abnormalities during isolation or sample contamination and groups were indicated with an appropriate n number.

Bearing in mind the 3R rule for the ethical use of animals in testing, we decided not to repeat the behavioral experiments performed previously by our group. In the previous studies, a behavioral pattern of changes occurring during OA progression was precisely described [24,54]. Both papers characterized OA-related pain behavior by Pressure Application Measurement test (Ugo Basile, Italy) and Dynamic Weight Bearing test (Bioseb, France). Additionally, these results were supported by microtomography–based 3-dimensional visualizations of rat knees in the consecutive days of the experiment [24], confirming permanent and irreversible changes within the studied subchondral bones of OA rats, correlated with disease progression.

### 4.2. Induction of Osteoarthritis

Rats were briefly anesthetized with 5% isoflurane (Forane®, Baxter Healthcare Corporation, USA) in 100% O2 (3 L/min). Joint damage was induced by a single intra-articular injection of sodium monoiodoacetate (MIA; 3 mg/50 μL; Sigma-Aldrich, Poznan, Poland) in 0.9% saline into the rear right knee. All surgical procedures were performed under sterile conditions in an animal procedure room away from animal holding areas.

### 4.3. RNA Extraction, cDNA Synthesis and Quantitative Real-Time Polymerase Chain Reaction

On days 2, 7, 14, 21 or 28 after MIA injection, the animals were sacrificed and a group of intact animals was used as a control. Control animals were sacrificed in various days (1 or 2 animals every experimental day) to minimize the differences associated with the duration of the experiment. No difference in the intact group in biochemical analysis was observed. The lumbar spinal cord (L4–L6 segments), cartilage and synovial membrane samples from the ipsilateral side were collected in RNA-later (Invitrogen) solution in individual tubes and stored at −80 °C until RNA isolation. Then, the samples were homogenized in 1 mL of Trizol reagent (Invitrogen, Carlsbad, CA, USA) and RNA isolation was performed according to the manufacturer’s protocol. The total RNA was quantified using a Nanodrop spectrophotometer (ND-1000, Nanodrop; Labtech International, UK). The samples were adjusted to a concentration of 1 μg μL^−1^ and reverse transcribed to cDNA using iScript reverse transcription supermix (Bio-Rad, Hercules, CA, USA) according to the manufacturer’s protocol. The qPCR reactions were carried out by iTaq universal probe supermix (Bio-Rad) and TaqMan assays (Thermo Fisher, Applied Biosystems, Waltham, MA, USA). The following assays were performed: Rn01527840_m1 (*Hprt1*), Rn00560865_m1 (*B2m*), Rn02758689_s1 (*Cnr1*), Rn04342831_s1 (*Cnr2*), Rn00583117_m1 (*Trpv1*), Rn01786262_m1 (*Nape-pld*), Rn00577086_m1 (*Faah*), Rn00668379_g1 (*Pla2g2a*), Rn01488539_m1 (*Abdh4*), Rn00583529_m1 (*Gde1*), Rn01514511_m1 (*Plcb1*), Rn01533758_m1 (*Ptpn22*), Rn01400935_m1 (*Inpp5d*), Rn01483828_m1 (*Ptgs2*), Rn01461082_ml (*Alox12*), Rn00696151_m1 (*Alox15*), Rn00560913 (*Camk2d*), Rn00578842_m1 (*Mapk14*), Rn00820922_g1 (*Mapk3*) and Rn00563784_m (*Atf3*). Reactions were run on a real-time PCR CFX96 touch system (Bio-Rad). Expression levels were assessed using the housekeeping genes *Hprt1* (lumbar spinal cord) or *B2m* (cartilage and synovial membrane). Cycle threshold values were calculated automatically by the CFX Manager software. RNA abundance was calculated as ddCT × 2^−(threshold cycle)^ and normalized to the reference gene values (*Hprt1* or *B2m*).

### 4.4. Statistical Analysis

The analysis was performed using Statistica 13 (StatSoft Software, Tulsa, OK, USA); graphs were prepared using Prism V.5 (GraphPad Software, La Jolla, CA, USA). All data are presented as the mean ± SEM. The results of RT-qPCR were evaluated by one-way analysis of variance (ANOVA) followed by Dunnett post hoc test. The groups included 39 animals. A value of *p* < 0.05 was considered to be statistically significant (* denotes *p* < 0.05; ** denotes *p* < 0.01; *** denotes *p* < 0.001 vs. intact animals).

## 5. Conclusions

In the present study, a detailed description of the role of ECS in the OA pathogenesis and progression has been provided. The data indicated that during OA progression, AEA synthesis and degradation enzymes were altered on the spinal and local levels. In addition to the main pathway, changes in the enzyme gene expression of two additional alternative pathways of AEA synthesis and degradation were observed. This result may indicate a role of ECS in OA development. Recent ECS-based therapies provide an expanded perspective on the development of an effective OA treatment; however, the detailed mechanism needs to be investigated. The presented study provides the missing information about the MIA-induced OA model and assists in the progress towards more adequate therapy in the future.

## Figures and Tables

**Figure 1 ijms-21-07381-f001:**
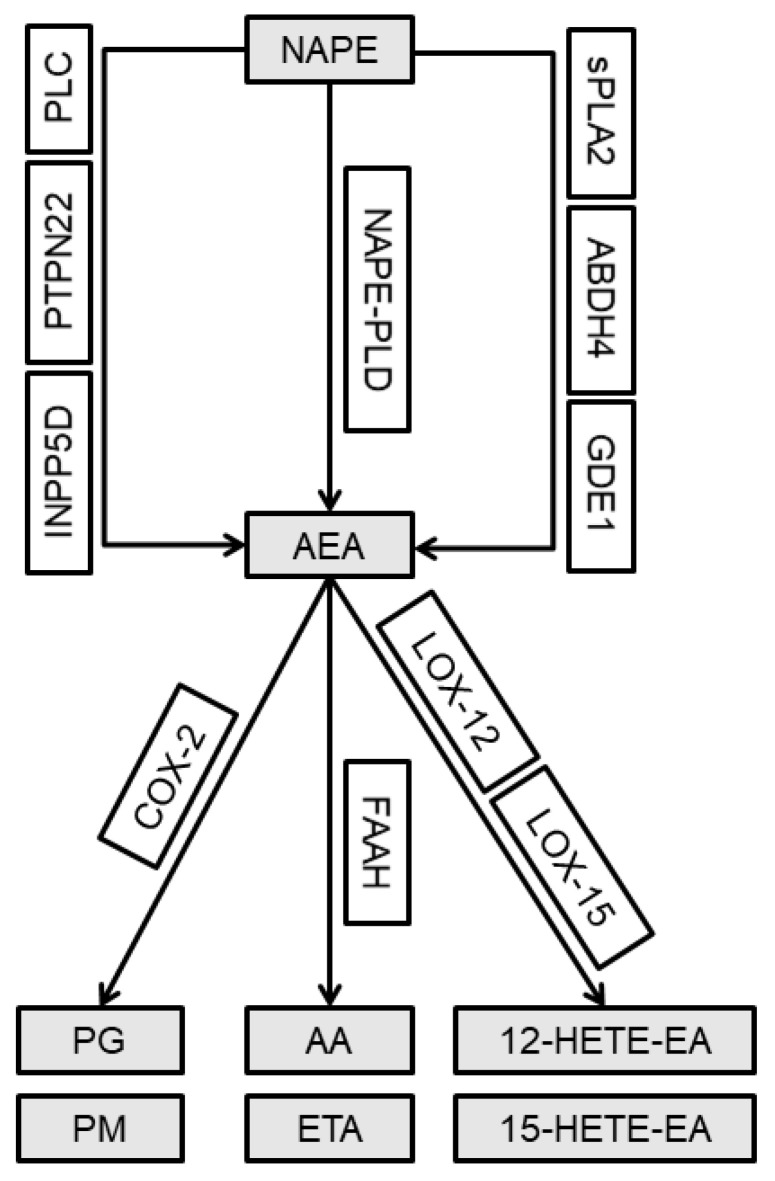
Simplified diagram of anandamide synthesis and degradation pathways. See text for details. Abbreviations: NAPE—*N*-arachidonoyl phosphatidylethanolamine; NAPE-PLD—*N*-arachidonoyl phosphatidylethanolamine phospholipase D; AEA—anandamide; PLC—phospholipase C; PTPN22—protein tyrosine phosphatase non-receptor type 22; INPP5D—phosphatidylinositol-3,4,5-trisphosphate 5-phosphatase 1; sPLA2—secreted phospholipase A2; ABDH4—β hydrolase domain-containing protein 4; GDE1—glycerophosphodiester phosphodiesterase 1; FAAH—fatty acid amide hydrolase; AA—arachidonic acid; ETA—ethanolamine; COX-2—cyclooxygenase 2; PG—prostaglandins; PM—prostamides; LOX-12/15—arachidonate lipoxygenases 12 and 15, respectively; 12/15-HETE-EA—12/15 hydroxyeicosatetraenoylethanolamide, respectively.

**Figure 2 ijms-21-07381-f002:**
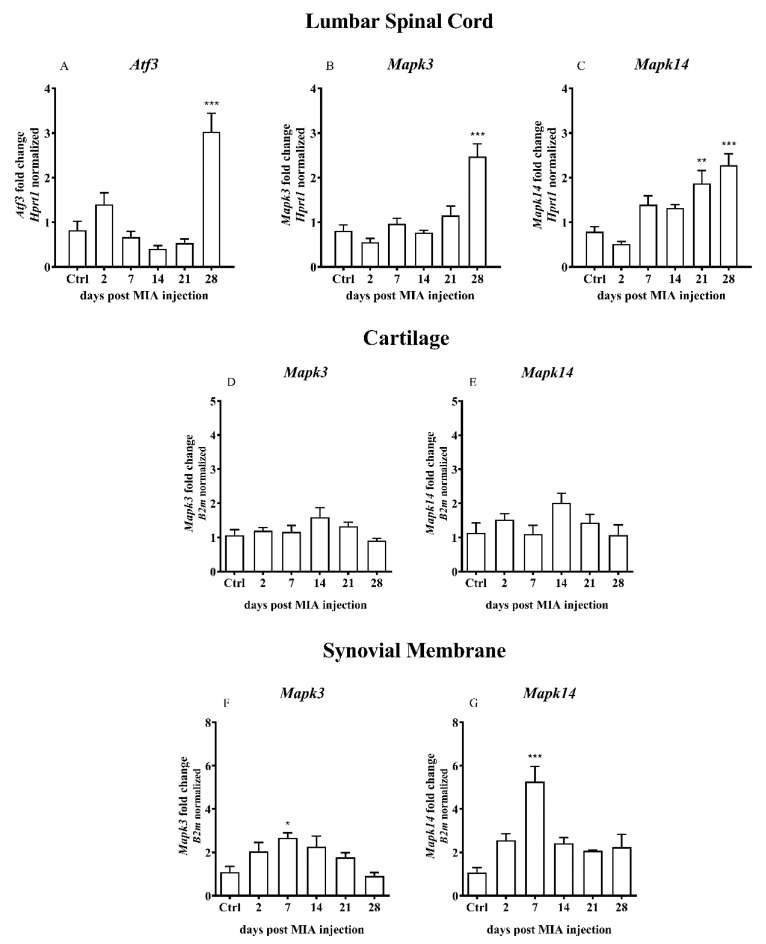
Transcript levels of cyclic AMP-dependent transcription factor (*Atf3*), mitogen-activated protein kinase 3 (*Mapk3*) and mitogen-activated protein kinase 14 (*Mapk14*) genes in the (**A**–**C**) lumbar L4–L6 spinal cord segments, (**D**,**E**) cartilage and (**F**,**G**) synovial membrane of rats after 3 mg MIA injection during OA development. The samples were collected 2, 7, 14, 21 and 28 days after MIA injection; the control group did not receive any treatment. Groups contained 6–9 samples (for the spinal cord analysis) or 3–6 (for the cartilage/synovial membrane analysis). Data are presented as the mean ± SEM of fold changes of the group average normalized versus the reference gene (*Hprt1* or *B2m*). Statistical analysis was performed using one-way ANOVA followed by Dunnett post hoc-test (intact animals treated as a control group). * denotes *p* < 0.05; ** denotes *p* < 0.01; *** denotes *p* < 0.001 vs. intact animals.

**Figure 3 ijms-21-07381-f003:**
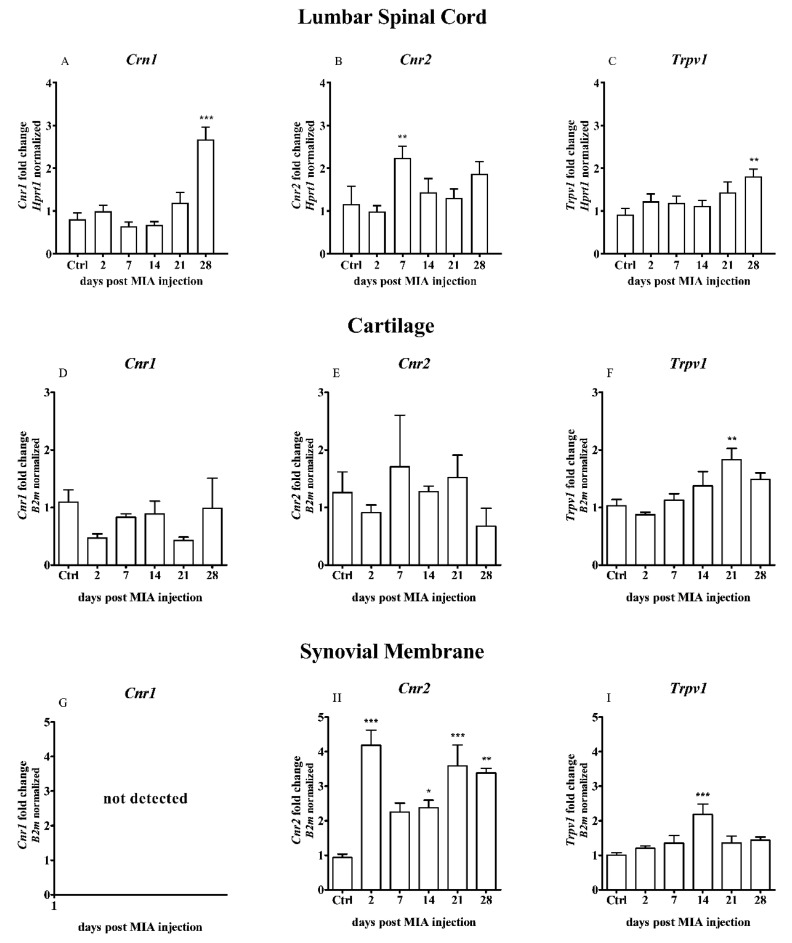
Transcript levels of the cannabinoid receptor type 1 and 2 (*Cnr1* and *Cnr2*) and transient receptor potential cation channel subfamily V member 1 (*Trpv1*) genes in the (**A**–**C**) lumbar L4-L6 spinal cord segments, (**D**–**F**) cartilage and (**G**–**I**) synovial membrane of rats after 3 mg MIA injection during OA development. Samples were collected 2, 7, 14, 21 and 28 days after MIA injection; the control group did not receive any treatment. Groups contained 6–9 samples (for the spinal cord analysis) or 3–6 samples (for the cartilage/synovial membrane analysis). Data are presented as the mean ± SEM of fold changes of the group average normalized versus the reference gene (*Hprt1* or *B2m*). Statistical analysis was performed using one-way ANOVA followed by Dunnett post hoc-test (intact animals treated as a control group). * denotes *p* < 0.05; ** denotes *p* < 0.01; *** denotes *p* < 0.001 vs. intact animals.

**Figure 4 ijms-21-07381-f004:**
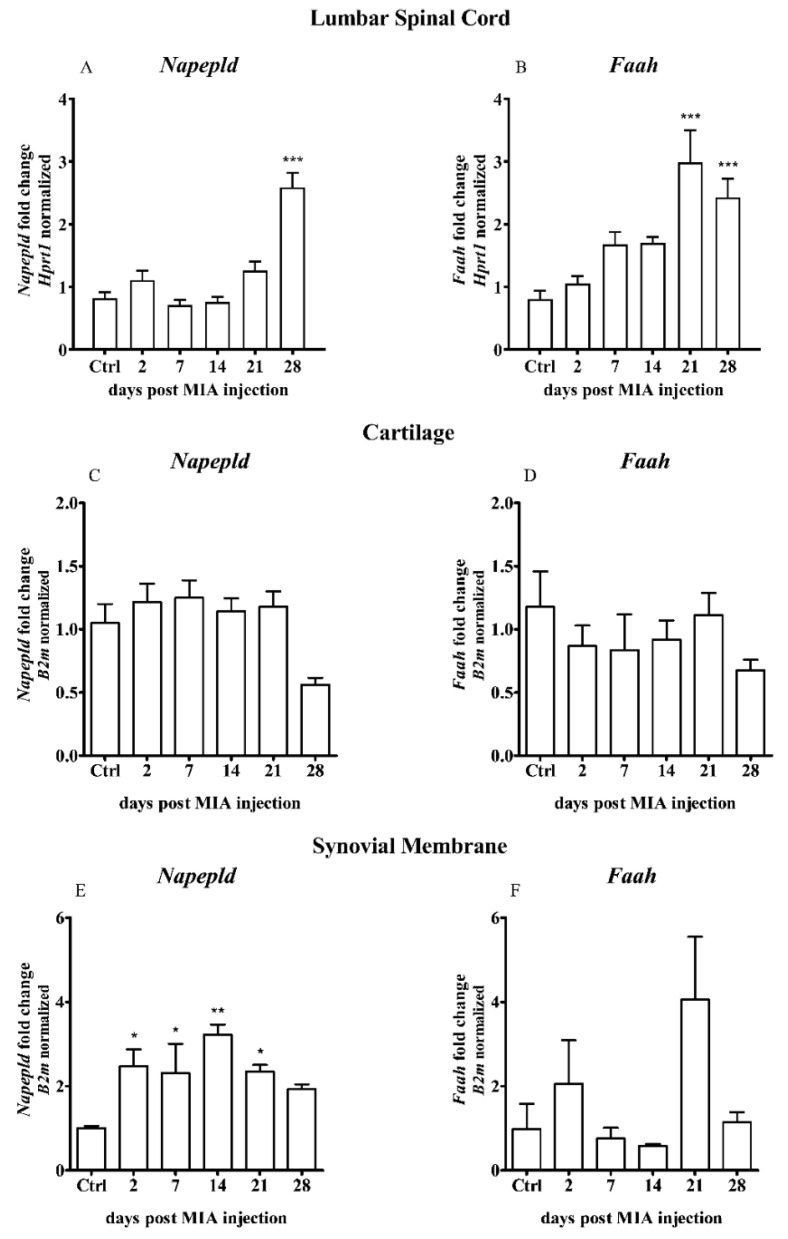
Transcript levels of the main enzymes of AEA synthesis and degradation, including N-acyl phosphatidylethanolamine-specific phospholipase D (*Nape-pld*) and fatty acid amide hydrolase (*Faah*) genes, in the (**A**,**B**) lumbar L4–L6 spinal cord segments, (**C**,**D**) cartilage and (**E**,**F**) synovial membrane of rats after 3 mg MIA injection during OA development. The samples were collected 2, 7, 14, 21 and 28 days after MIA injection; the control group did not receive any treatment. The groups contained 6–9 samples (for the spinal cord analysis) or 3–6 samples (for the cartilage/synovial membrane analysis). Data are presented as the mean ± SEM of fold changes of the group average normalized versus the reference gene (*Hprt1* or *B2m*). Statistical analysis was performed using one-way ANOVA followed by Dunnett post hoc-test (intact animals treated as a control group). * denotes *p* < 0.05; ** denotes *p* < 0.01; *** denotes *p* < 0.001 vs. intact animals.

**Figure 5 ijms-21-07381-f005:**
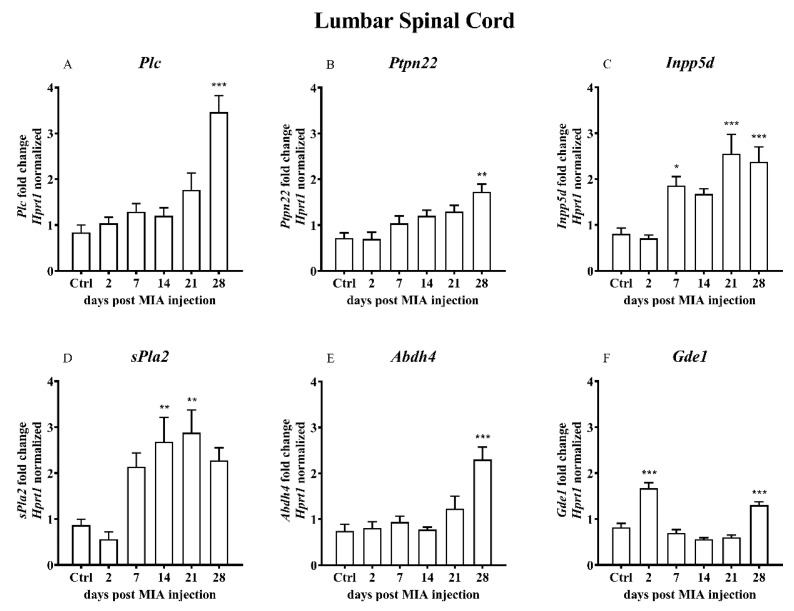
Transcript levels of the alternative AEA synthesis and degradation pathways (**A**–**I**), including phospholipase C (*Plc*), protein tyrosine phosphatase, non-receptor type 22 (*Ptpn22*), inositol polyphosphate-5-phosphatase D (*Inpp5d*), phospholipase A2 (*sPla2g2a*), β hydrolase domain containing protein 4 (*Abdh4*), glycerophosphodiester phosphodiesterase 1 (*Gde1*), prostaglandin-endoperoxide synthase 2 (*Ptgs2*), arachidonate 12-lipoxygenase (*Alox12*) and arachidonate 15-lipoxygenase (*Alox15*), in the lumbar L4–L6 spinal cord segments of rats after 3 mg MIA injection during OA development. Samples were collected 2, 7, 14, 21 and 28 days after MIA injection; the control group did not receive any treatment. The groups contained 6–9 samples. Data are presented as the mean ± SEM of fold changes of the group average normalized versus the reference gene (*Hprt1*). Statistical analysis was performed using one-way ANOVA followed by Dunnett post hoc-test (intact animals treated as a control group). * denotes *p* < 0.05; ** denotes *p* < 0.01; *** denotes *p* < 0.001 vs. intact animals.

**Figure 6 ijms-21-07381-f006:**
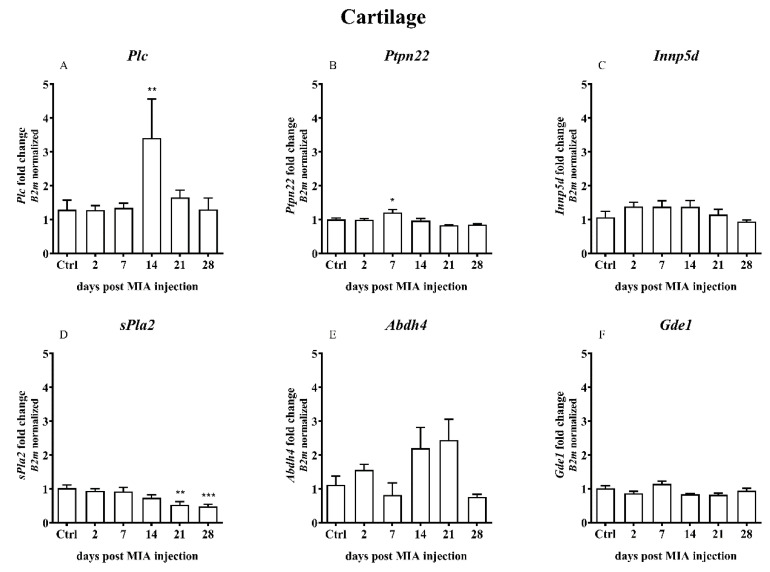
Transcript levels of the alternative AEA synthesis and degradation pathways (**A**–**I**), including phospholipase C (*Plc*), protein tyrosine phosphatase, non-receptor type 22 (*Ptpn22*), inositol polyphosphate-5-phosphatase D (*Inpp5d*), phospholipase A2 (*sPla2g2a*), β hydrolase domain containing protein 4 (*Abdh4*), glycerophosphodiester phosphodiesterase 1 (*Gde1*), prostaglandin-endoperoxide synthase 2 (*Ptgs2*), arachidonate 12-lipoxygenase (*Alox12*) and arachidonate 15-lipoxygenase (*Alox15*), in the cartilage of rats after 3 mg MIA injection during OA development. The samples were collected 2, 7, 14, 21 and 28 days after MIA injection; the control group did not receive any treatment. The groups contained 3 - 6 samples. Data are presented as the mean ± SEM of fold changes of the group average normalized versus the reference gene (*B2m*). Statistical analysis was performed using one-way ANOVA followed by Dunnett post hoc-test (intact animals treated as a control group). * denotes *p* < 0.05; ** denotes *p* < 0.01; *** denotes *p* < 0.001 vs. intact animals.

**Figure 7 ijms-21-07381-f007:**
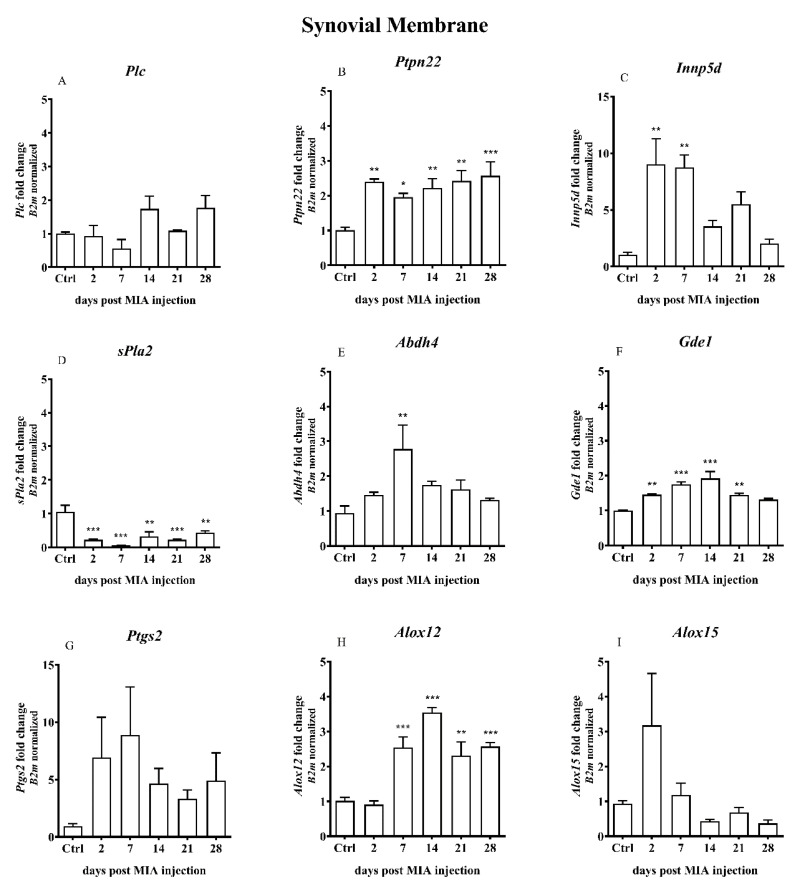
Transcript levels of the alternative AEA synthesis and degradation pathways (**A**–**I**), including phospholipase C (*Plc*), protein tyrosine phosphatase, non-receptor type 22 (*Ptpn22*), inositol polyphosphate-5-phosphatase D (*Inpp5d*), phospholipase A2 (*sPla2g2a*), β hydrolase domain containing protein 4 (*Abdh4*), glycerophosphodiester phosphodiesterase 1 (*Gde1*), prostaglandin-endoperoxide synthase 2 (*Ptgs2*), arachidonate 12-lipoxygenase (*Alox12*) and arachidonate 15-lipoxygenase (*Alox15*), in the synovial membrane of rats after 3 mg MIA injection during OA development. The samples were collected 2, 7, 14, 21 and 28 days after MIA injection; the control group did not receive any treatment. The groups contained 3–6 samples. Data are presented as the mean ± SEM of fold changes of the group average normalized versus the reference gene (*B2m*). Statistical analysis was performed using one-way ANOVA followed by Dunnett post hoc-test (intact animals treated as a control group). * denotes *p* < 0.05; ** denotes *p* < 0.01; *** denotes *p* < 0.001 vs. intact animals.

**Figure 8 ijms-21-07381-f008:**
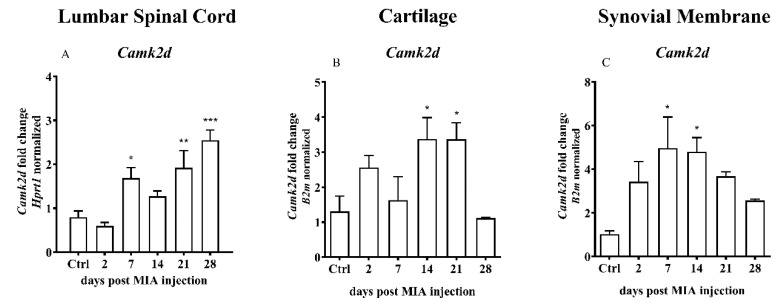
Transcript levels of calcium/calmodulin-dependent protein kinase II delta (*Camk2d*) in the (**A**) lumbar L4–L6 spinal cord segments, (**B**) cartilage and (**C**) synovial membrane of rats after 3 mg MIA injection during OA development. The samples were collected 2, 7, 14, 21 and 28 days after MIA injection; the control group did not receive any treatment. The groups contained 6–9 samples (for the spinal cord analysis) or 3 - 6 samples (for the cartilage/synovial membrane analysis). Data are presented as the mean ± SEM of fold changes of the group average normalized versus the reference gene (*Hprt1* or *B2m*). Statistical analysis was performed using one-way ANOVA followed by Dunnett post hoc-test (intact animals treated as a control group). * denotes *p* < 0.05; ** denotes *p* < 0.01; *** denotes *p* < 0.001 vs. intact animals.

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
