# Peer review of "Alterations in Anandamide Synthesis and Degradation during Osteoarthritis Progression in an Animal Model"

_ijms, 2020, doi:10.3390/ijms21197381_

Round 1

Reviewer 1 Report

In the present study gene expression alterations of various components of the endocannabinoid system (ECS) were investigated in a rat osteoarthritis model.

Modest changes of various gene expressions were demonstrated in the lumbar spinal cord, cartilage and synovial membrane. The authors reach very broad conclusions based on their data, claiming e.g. that “detailed description of the role of ECS in the OA pathogenesis and progression has been provided” or “connection between MIA-induced osteoarthritic pain and central nerve sensitization was demonstrated” or that “the data of the present study indicate that NAPE-PLD (the main enzyme responsible for AEA 321 synthesis) plays a role during chronic pain formation”.

The topic is worthy to investigate, however there are some important flaws that must be addressed. While endocannabinoids may indeed play a prominent role in painful conditions, the presented data do not consistently provide evidence to support the abovementioned statements. Firstly, the time course of gene expression changes do not show a pattern which could be identified as a certain phase of the disease progression. In fact, since the authors did not assess pain severity, only histological arthritis severity score, therefore direct comparisons of gene alterations and OA pain progression cannot be made. Only literature data can be used to make sense of the temporal pattern of transcriptomic changes. Secondly, even if we saw a consistent pattern, a causal link between a gene and the pain cannot be claimed based on the described gene expression changes. Lastly, some of the statistically significant gene expression changes seem biologically insignificant.

Another major weakness is the study design. There is no parallel control for all time points, but only one control sample and it is not clear when these control samples were taken (day 0 or 28?). Therefore it cannot be excluded that some of the statistically significant differences might be temporal fluctuations due to factors other than the OA.

All these issues are fundamental limitations of the study which need to be adequately addressed in the Discussion. The paper must be revised accordingly.

Author Response

Dr Rosaria Meccariello

Guest Editor

Special Issue "Endocannabinoid System in Health and Disease: Current Situation and Future Perspectives 2.0"

September 28, 2020

Dear Professor Meccariello,

Thank you very much for your letter dated 20 Sept 2020 and the enclosed reviewers’ comments concerning our manuscript ijms-938987 entitled "Alterations in anandamide synthesis and degradation during osteoarthritis progression in an animal model". We have improved the paper according to the suggestions and responded adequately to the Reviewers’ comments. The changes in the revised manuscript are discussed and are enclosed in two separate files ‘‘Response to Reviewer 1 Comments Manuscript ijms-938987”and ‘‘Response to Reviewer 2 Comments Manuscript ijms-938987”.

We appreciate the valuable comments made by the Reviewers concerning our paper. We've implemented all of the suggestion including adapting the manuscript to a brief research report standard and resubmitted the final version of the manuscript to the journal. We hope that the corrected and improved thanks to these critical remarks version of our paper could now be accepted for publication.

Q1: In the present study gene expression alterations of various components of the endocannabinoid system (ECS) were investigated in a rat osteoarthritis model. Modest changes of various gene expressions were demonstrated in the lumbar spinal cord, cartilage and synovial membrane. The authors reach very broad conclusions based on their data, claiming e.g. that “detailed description of the role of ECS in the OA pathogenesis and progression has been provided” or “connection between MIA-induced osteoarthritic pain and central nerve sensitization was demonstrated” or that “the data of the present study indicate that NAPE-PLD (the main enzyme responsible for AEA 321 synthesis) plays a role during chronic pain formation”. The topic is worthy to investigate, however there are some important flaws that must be addressed. While endocannabinoids may indeed play a prominent role in painful conditions, the presented data do not consistently provide evidence to support the abovementioned statements. Firstly, the time course of gene expression changes do not show a pattern which could be identified as a certain phase of the disease progression. In fact, since the authors did not assess pain severity, only histological arthritis severity score, therefore direct comparisons of gene alterations and OA pain progression cannot be made. Only literature data can be used to make sense of the temporal pattern of transcriptomic changes.

Bearing in mind the 3R rule for the ethical use of animals in testing, we decided not to repeat the behavioural experiments performed previously by our group. In the previous studies, a behavioural pattern of changes occurring during OA progression was precisely described (Malek et al. 2015; Pajak et al. 2017). Both papers characterized OA-related pain behaviour by Pressure Application Measurement test (Ugo Basile, Italy) and Dynamic Weight Bearing test (Bioseb, France). Experiments described there were performed on the same species (male Wistar rats) using the same MIA dose (3 mg) for OA induction, as well as in the same experimental conditions (e.g. switching lights on and off, indoor temperature and humidity). Both studies revealed a significant decrease in pain threshold observed already on the 2nd day of the experiment, with a similar characteristic drop of pain threshold in the initial phase of OA (associated with a transient local inflammatory state, induced by intra-articular MIA injection) and a stable, persisted pain threshold decrease after 10 days post-MIA treatment. Additionally, these results were supported by microtomography–based 3-dimensional visualizations of rat knees in the consecutive days of the experiment (Malek et al. 2015), confirming permanent and irreversible changes within the studied subchondral bones of OA rats, correlated with disease progression. Moreover, several other studies describe behavioural pain alterations in the MIA-induced OA rat model (Y. M. Lee et al. 2019; Philpott, O’Brien, and McDougall 2017; S. Y. Lee et al. 2018). All of them are in line with the results of our group and show a sustained pain elevation several days post-MIA injection up to 14-29 days post treatment, depending on the experimental design.

For the above reasons, we did not repeat the behavioural experiments. Relevant explanations have been included in the revised version of manuscript (lines 446-453).

Q2: Secondly, even if we saw a consistent pattern, a causal link between a gene and the pain cannot be claimed based on the described gene expression changes.

In the current study we only suggest the mechanism of pathological changes during OA progression, we cannot state and we cannot and would not undertake to present a thesis indicating the link between gene expression and pain. However, we hypothesize that  gene expressions paralleling pain behaviour may indicate the relationship of this gene with the development of the disease.

Q3: Lastly, some of the statistically significant gene expression changes seem biologically insignificant.

In line with this Reviewer's remark, we have edited the discussion, focusing on those changes that may be of greater significance for the potential future therapies. It is difficult to specify all modified sentences, where amendments were applied. We believe we have addressed Reviewer’s comment correctly. For details, please see the revised discussion (highlited using “track changes” function in the Microsoft Word file).

Q4: Another major weakness is the study design. There is no parallel control for all time points, but only one control sample and it is not clear when these control samples were taken (day 0 or 28?). Therefore it cannot be excluded that some of the statistically significant differences might be temporal fluctuations due to factors other than the OA.

Always when planning the experiment, we try to secure the maximum number of control groups (of course, this does not exclude that we could forget about something). However, in this case, our shortcoming is the fact, that despite a correctly planned experiment, it was not accompanied by an accurate and reflecting description, due to the imprecise description of the methodology, it was not specified in the manuscript. To present the matter as it was and to address the valuable reviewer’s comment the following sentences are added to the manuscript (lines 462-464): “Control animals were sacrificed in various days (1 or 2 animals every experimental day) to minimize the differences associated with the duration of the experiment. No difference in the intact group in biochemical analysis was observed.”

Q5: All these issues are fundamental limitations of the study which need to be adequately addressed in the Discussion. The paper must be revised accordingly.

We believe, that all out clarifications and comments would increase the value of the manuscript and would satisfy the Reviewer.

We are looking forward to your reply.

Yours sincerely,

Dr Katarzyna Starowicz, PhD DSc

Maj Institute of Pharmacology Polish Academy of Sciences

12 Smetna str, 31-343 Krakow Poland

Phone: +48 12 6623206

Email: starow@if-pan.krakow.pl

References:

Lee, Seon Yeong, Seung Hoon Lee, Hyun Sik Na, Ji Ye Kwon, Goo Young Kim, Kyung Ah Jung, Keun Hyung Cho, et al. 2018. “The Therapeutic Effect of STAT3 Signaling-Suppressed MSC on Pain and Articular Cartilage Damage in a Rat Model of Monosodium Iodoacetate-Induced Osteoarthritis.” Frontiers in Immunology. https://doi.org/10.3389/fimmu.2018.02881.

Lee, Yun Mi, Eunjung Son, Seung Hyung Kim, and Dong Seon Kim. 2019. “Effect of Alpinia Oxyphylla Extract in Vitro and in a Monosodium Iodoacetate-Induced Osteoarthritis Rat Model.” Phytomedicine. https://doi.org/10.1016/j.phymed.2019.153095.

Malek, Natalia, Monika Mrugala, Wioletta Makuch, Natalia Kolosowska, Barbara Przewlocka, Marcin Binkowski, Martyna Czaja, Enrico Morera, Vincenzo Di Marzo, and Katarzyna Starowicz. 2015. “A Multi-Target Approach for Pain Treatment: Dual Inhibition of Fatty Acid Amide Hydrolase and TRPV1 in a Rat Model of Osteoarthritis.” Pain156 (5): 890–903. https://doi.org/10.1097/j.pain.0000000000000132.

Pajak, Agnieszka, Magdalena Kostrzewa, Natalia Malek, Michal Korostynski, and Katarzyna Starowicz. 2017. “Expression of Matrix Metalloproteinases and Components of the Endocannabinoid System in the Knee Joint Are Associated with Biphasic Pain Progression in a Rat Model of Osteoarthritis.” Journal of Pain Research. https://doi.org/10.2147/JPR.S132682.

Philpott, Holly T., Melissa O’Brien, and Jason J. McDougall. 2017. “Attenuation of Early Phase Inflammation by Cannabidiol Prevents Pain and Nerve Damage in Rat Osteoarthritis.” Pain 158 (12): 2442–51. https://doi.org/10.1097/j.pain.0000000000001052.

Reviewer 2 Report

The manuscript is interested and is well written. I suggest to expand disscussion including problem what significance for development of signs of OA and pain could have changes in alternative  AEA synthesis and degradation pathways taking into account the final products of these pathways. Considering the possibility of therapy with cannabinoids I suggest to add some information concerning anti-inflammatory action of cannabinoids.

Author Response

Dr Rosaria Meccariello

Guest Editor

Special Issue "Endocannabinoid System in Health and Disease: Current Situation and Future Perspectives 2.0"

September 28, 2020

Dear Professor Meccariello,

Thank you very much for your letter dated 20 Sept 2020 and the enclosed reviewers’ comments concerning our manuscript ijms-938987 entitled "Alterations in anandamide synthesis and degradation during osteoarthritis progression in an animal model". We have improved the paper according to the suggestions and responded adequately to the Reviewers’ comments. The changes in the revised manuscript are discussed and are enclosed in two separate files ‘‘Response to Reviewer 1 Comments Manuscript ijms-938987” and ‘‘Response to Reviewer 2 Comments Manuscript ijms-938987”.

We appreciate the valuable comments made by the Reviewers concerning our paper. We've implemented all of the suggestion including adapting the manuscript to a brief research report standard and resubmitted the final version of the manuscript to the journal. We hope that the corrected and improved thanks to these critical remarks version of our paper could now be accepted for publication.

Q1: The manuscript is interested and is well written. I suggest to expand discussion including problem what significance for development of signs of OA and pain could have changes in alternative AEA synthesis and degradation pathways taking into account the final products of these pathways.

To correctly address this reviewer’s comment and to support the importance of alternative AEA’s metabolism pathways the following sentences are added to the manuscript (lines 406-423):

“In the current study we proved the AEA synthesis and degradation enzyme upregulation, during the course of OA. Increase in the AEA synthesis enzymes lead to an increase in the AEA level, which is a desired result, because of the AEA’s analgesic effect. In turn, AEA degradation enzymes’ enhancement increase the level of AEA metabolites (arachidonic acid, ethanolamine, prostaglandins, prostamides, 12-/15-HETE-EA) and reduce the level of AEA. The latter AEA’s metabolites, derived on the main and alternative pathways can be involved in the inflammatory process. Therefore an effective approach to omit problem might be dual-acting substances, e.g. FAAH/COX-2 inhibitors, that target both enzymes. Dual-acting drugs offer an analgesic effect by elevating the endogenously produced endocannabinoids (by inhibiting FAAH) and lower the production of pro-inflammatory prostaglandins (by inhibiting COX-2) (summarized in Malek and Starowicz 2016). In turn, LOX-12/15 metabolites may act in an analgesic way. Indeed, in an animal model of neuropathic pain (chronic constriction injury, CCI, to the sciatic nerve), FAAH inhibitor URB597, diminished thermal and tactile allodynia, but also decreased the spinal AEA level and increased LOX-15 level at the same time. This may lead to the TRPV1-mediated analgesia in CCI rats, via 15-hydroxy-AEA, together with oleoylethanolamide and palmitoylethanolamide (Starowicz et al. 2013). AEA metabolites can also be important in several other pathologies. Turcotte et al. widely summarizes the regulatory role of AEA metabolites in various diseases (Turcotte et al. 2015). Nevertheless, to confirm this hypothesis the levels of metabolites should be measured in the animals’ tissues, what is a proper direction for the future research.”

Q2: Considering the possibility of therapy with cannabinoids I suggest to add some information concerning anti-inflammatory action of cannabinoids.

The following sentences are added to the manuscript (lines 424-435):

”Considering the analgesic and anti-inflammatory effects of cannabinoids in the pre-clinical studies, cannabinoid therapy seems to be a promising target for the treatment of several diseases. In the arthritis animal models, phytocannabinoid cannabidiol (CBD) reduced inflammation and analgesia in a rat model (Hammell et al. 2016; Philpott, O’Brien, and McDougall 2017). CBD may also preferentially target inflammatory-activated fibroblasts and reduce its viability, therefore may have an anti-arthritic activity (Lowin et al. 2020). Synthetic CB2 receptor agonists were also proven to exert an anti-inflammatory response in several arthritis animal models. JWH-015 inhibited inflammation in the rheumatoid arthritis synovial fibroblasts cell cultures and in the arthritis rat model (Fechtner et al. 2019). JWH133 suppressed collagen-induced arthritis in mice, acted anti-inflammatory by repolarizing macrophages from the M1 to M2 phenotype and reduced pro-inflammatory cytokine expression (Zhu et al. 2019). CB2 agonist 4Q3C showed an anti-inflammatory effect in rheumatoid arthritis mouse model (reduced bone erosion, inhibited formation of osteoclasts and lowered the level of TNFα, IL-1β, COX-2 and inducible NO synthase) (Bai et al. 2019).”

We believe, that all out clarifications and comments would increase the value of the manuscript and would satisfy the Reviewer.

We are looking forward to your reply.

Yours sincerely,

Dr Katarzyna Starowicz, PhD DSc

Maj Institute of Pharmacology Polish Academy of Sciences

12 Smetna str, 31-343 Krakow Poland

Phone: +48 12 6623206

Email: starow@if-pan.krakow.pl

References:

Bai, Jiaxiang, Gaoran Ge, Yijun Wang, Wenhao Zhang, Qing Wang, Wei Wang, Xiaobin Guo, et al. 2019. “A Selective CB2 Agonist Protects against the Inflammatory Response and Joint Destruction in Collagen-Induced Arthritis Mice.” Biomedicine and Pharmacotherapy. https://doi.org/10.1016/j.biopha.2019.109025.

Fechtner, Sabrina, Anil K. Singh, Ila Srivastava, Christopher T. Szlenk, Tim R. Muench, Senthil Natesan, and Salahuddin Ahmed. 2019. “Cannabinoid Receptor 2 Agonist JWH-015 Inhibits Interleukin-1βinduced Inflammation in Rheumatoid Arthritis Synovial Fibroblasts and in Adjuvant Induced Arthritis Rat via Glucocorticoid Receptor.” Frontiers in Immunology. https://doi.org/10.3389/fimmu.2019.01027.

Hammell, D. C., L. P. Zhang, F. Ma, S. M. Abshire, S. L. McIlwrath, A. L. Stinchcomb, and K. N. Westlund. 2016. “Transdermal Cannabidiol Reduces Inflammation and Pain-Related Behaviours in a Rat Model of Arthritis.” European Journal of Pain (United Kingdom) 20 (6): 936–48. https://doi.org/10.1002/ejp.818.

Lowin, Torsten, Ren Tingting, Julia Zurmahr, Tim Classen, Matthias Schneider, and Georg Pongratz. 2020. “Cannabidiol (CBD): A Killer for Inflammatory Rheumatoid Arthritis Synovial Fibroblasts.” Cell Death and Disease 11 (714). https://doi.org/10.1038/s41419-020-02892-1.

Malek, Natalia, and Katarzyna Starowicz. 2016. “Dual-Acting Compounds Targeting Endocannabinoid and Endovanilloid Systems-a Novel Treatment Option for Chronic Pain Management.” Frontiers in Pharmacology. https://doi.org/10.3389/fphar.2016.00257.

Philpott, Holly T., Melissa O’Brien, and Jason J. McDougall. 2017. “Attenuation of Early Phase Inflammation by Cannabidiol Prevents Pain and Nerve Damage in Rat Osteoarthritis.” Pain 158 (12): 2442–51. https://doi.org/10.1097/j.pain.0000000000001052.

Starowicz, Katarzyna, Wioletta Makuch, Michal Korostynski, Natalia Malek, Michal Slezak, Magdalena Zychowska, Stefania Petrosino, et al. 2013. “Full Inhibition of Spinal FAAH Leads to TRPV1-Mediated Analgesic Effects in Neuropathic Rats and Possible Lipoxygenase-Mediated Remodeling of Anandamide Metabolism.” PLoS ONE 8 (4). https://doi.org/10.1371/journal.pone.0060040.

Turcotte, Caroline, Francois Chouinard, Julie S. Lefebvre, and Nicolas Flamand. 2015. “Regulation of Inflammation by Cannabinoids, the Endocannabinoids 2-Arachidonoyl-Glycerol and Arachidonoyl-Ethanolamide, and Their Metabolites.” Journal of Leukocyte Biology. https://doi.org/10.1189/jlb.3ru0115-021r.

Zhu, Mo, Binqin Yu, Jiaxiang Bai, Ximing Wang, Xiaobin Guo, Yu Liu, Jiayi Lin, et al. 2019. “Cannabinoid Receptor 2 Agonist Prevents Local and Systemic Inflammatory Bone Destruction in Rheumatoid Arthritis.” Journal of Bone and Mineral Research. https://doi.org/10.1002/jbmr.3637.

Round 2

Reviewer 1 Report

Thank you for the revision, I have no further comments.